# Exploring Explicit Representations in 4D: A Comparative Analysis with HexPlane

## Abstract

Modeling and re-rendering novel views of dynamic 3D scenes is a challenging problem in 3D vision. Employing implicit representations for the task, extending static NeRFs to 4D incurs high computational costs due to the numerous MLP evaluations, highlighting the need for efficient representations of dynamic 3D scenes. Non-Nerf Methods such as Niemeyer et al. (2019), Jiang et al. (2022), and Jiang et al. (2021) have primarily been applied to idealized, single-subject scenes and have not yet been adapted for real-world camera images. Cao and Johnson (2023) proposes using HexPlane, an explicit scene representation method that factors a 4D volume into six feature planes. This paper attempts to verify their claims and compare them with similar methods like Gaussian Splatting by Wu et al. (2023) and K-planes by Fridovich-Keil et al. (2023). We conduct a thorough examination of the architectural choices and design elements inherent in HexPlane and further incorporate additional regularization to achieve a performance improvement.

## 1 Introduction

Reconstructing and re-rendering 3D scenes from a set of 2D images is a core vision problem that can enable numerous AR/VR applications and amplify the field's horizons. Significant advancements have been achieved in reconstructing static scenes, but as we know, the real world is dynamic, and motion is the norm. Recent works have started exploring this demanding problem of dynamic scene reconstruction.

Current methods of reconstructing dynamic scenes to address the core vision problem of re-rendering 3D scenes from 2D images can be categorized into NeRF-based and Non-NeRF-based. Most recent works build upon Neural Radiance Fields (NeRF) Mildenhall et al. (2020), utilizing implicit scene representations. They train a large multi-layer perceptron (MLP) that takes as input the position of a point in space and time and outputs either the point's color or deformation to a canonical static scene. In either case, rendering images from novel views is expensive since each generated pixel requires many MLP evaluations. Similarly, training is also slow and computationally expensive, limiting the possibility of real-time application of these methods.

Non-NeRF-based representation methods like Niemeyer et al. (2019), Jiang et al. (2022), and Jiang et al. (2021) were primarily tested on synthetic single-subject scenes with defined structures, some specifically designed for human structures, unlike NeRF-based methods which have proven to handle real world, multi-subject, crowded scenes captured using camera images. Hence, we lean on NeRF-based methods to stay more relevant to current research demands and cater to real-world camera images.

Cao and Johnson (2023) proposes a novel method for explicitly representing dynamic scenes, HexPlane, building upon Müller et al. (2022), Chen et al. (2022), which employ similar methods on static scenes. The Hexplane authors have designed a spatial-temporal data structure that stores scene data. HexPlane decomposes a 4D spacetime grid into six feature planes spanning each pair of coordinate axes (e.g., XY, ZT). The fused feature vector is then passed to a tiny MLP that predicts the point's color; novel views can be rendered via volume rendering. They claim to achieve 100x speed-ups on prior methods tackling this problem.

HexPlane is increasingly adopted as a fundamental baseline in studies involving explicit representation in 4D spaces. Given the rising prominence of Hexplane, particularly with subsequent methodologies like 4D Gaussian Splatting building upon it, we aim to comprehensively verify Hexplane's components and architectural decisions, which are crucial for ensuring its reliability as a baseline.

To this end, we contribute the following :

1. Successfully reproduced the work and verified all the claims made by Cao and Johnson (2023) by thoroughly assessing the choice of architectural and design elements.

2. Demonstrated HexPlane's robustness by experimenting with new datasets and benchmarking against methods like Gaussian Splatting by Wu et al. (2023) and K-planes by Fridovich-Keil et al. (2023)

3. Achieved minor performance improvement by integrating temporal smoothness regularization.

## 2 Background and Related Work

### 2.1 Neural Scene Representation

NeRF and its variants Barron et al. (2021a), Barron et al. (2021b), employing neural networks to represent 3D scenes implicitly have shown impressive results on novel view synthesis and related fields in the 3D vision space. Many recent papers propose hybrid representations that combine a fast explicit scene representation with learnable neural network components to address the challenge of costly implicit neural representations, providing significant speedups over purely implicit methods. Various explicit representation methods have been studied, such as Huang et al. (2023), Yu et al. (2021), Chen et al. (2021), but they assume a static 3D scene.

Inspired by the quality of results achieved by implicit scene representation methods on static scenes, Park et al. (2021a) and Park et al. (2021b) have expanded the boundaries of novel view synthesis for dynamic scenes. One line of research represents dynamic scenes by extending NeRF with an additional time dimension (T-NeRF) Gao et al. (2021a) or additional latent code. Despite the ability to represent general topology changes, they suffer from a severely under-constrained problem, requiring additional supervision like depths, optical flows, or dense observations for decent results. Research has been conducted using an explicit voxel grid to model temporal information, substantially accelerating the learning time for dynamic scenes. Methods like Shao et al. (2023) and Wang et al. (2023) represent further advancements in faster dynamic scene learning by adopting decomposed neural voxels. They treat sampled points in each timestamp individually. Though these methods achieve fast training, real-time rendering for dynamic scenes is still challenging, especially for monocular input.

### 2.2 Accelerating NeRF

Multiple works have been proposed to accelerate NeRFs at diverse stages by improving the inference speed by optimizing the computation Fang et al. (2022a) or by reducing the training times by learning a generalizable model Chen et al. (2021). Recently, substantial improvements have been observed in both training and rendering durations by employing a hybrid model.

Several studies, such as Fridovich-Keil et al. (2022), have employed geometric representations to notably reduce optimization times by using trilinear interpolation in a 3D grid. However, the scalability of these explicit grid structures, similar to Sun et al. (2022), is limited as they expand exponentially with increasing dimensions, posing challenges for high-resolution and 4D dynamic volumes.

Müller et al. (2022) introduced a compact, multiresolution voxel grid encoded implicitly through a hash function, enabling rapid optimization and rendering. Similarly, Chen et al. (2022) enhanced speed and compressed models by substituting the voxel grid with a tensor decomposition of planes and vectors. In a generative context, Chan et al. (2021) implemented a spatial decomposition involving three planes, combining their values to represent a 3D volume.

Extensions of NeRF to accommodate dynamic scenes typically follow one of two approaches: (1) overlaying a deformation field on a static canonical field [Du et al. (2021), Fang et al. (2022b), Li et al. (2021), Park et al. (2021a), Pumarola et al. (2020), Tretschk et al. (2021), Yuan et al. (2021)], which simplifies separating static and dynamic elements but struggles with topological changes, or (2) learning a time-conditioned radiance field [Gao et al. (2021b), Li et al. (2022), Li et al. (2021), Park et al. (2021b), Xian et al. (2021)], which complicates the separation of static and dynamic components. A third approach involves repeating a 3D space representation at each timestep, as seen in Song et al. (2023). This can lead to overly large models that fail to account for space-time interactions in lengthy videos.

In line with these ideas, HexPlane employs an explicit dynamic scene representation by factoring a 4D spatial-temporal space into six feature planes and then using a tiny MLP at the end to decode the color and opacity associated with the voxels.

### 2.3 Hexplane

### 2.3.1 Architecture

This paper aims to replicate the Hexplane approach of rendering novel views in dynamic 3D scenes, leveraging a hybrid model that combines an explicit representation of the scene with a compact Multilayer Perceptron (MLP) for only the decoder mechanism, as shown in figure 1. Following the paradigm established by NeRF, the HexPlane model predicts color and opacity for points in spacetime, facilitating image rendering from novel vantage points and moments through differentiable volumetric rendering.

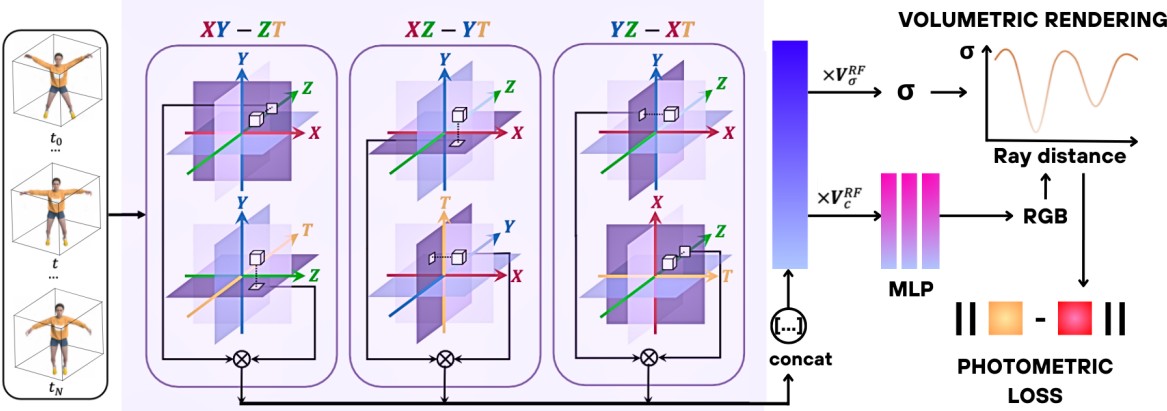

Figure 1: Method Overview: HexPlane has six feature planes, each spanning coordinate axes pairs (e.g., XY, ZT). Point features are computed by multiplying vectors from paired planes, then concatenated and multiplied by VRF. A compact MLP predicts RGB colors. Training is done via photometric loss minimization between rendered and target images. (Cao and Johnson, 2023)

### 2.3.2 Factorization and Temporal Modelling

HexPlane proposes a factorization technique that addresses the issue of memory consumption in naïve representations, where a 4D volume is represented as independent static 3D per time step. The standard method used in some previous works, such as Chan et al. (2022), lacks information sharing across frames, which is critical to counter the issue of sparse observations [Equation 1]. Volume Basis factorization enables information sharing by representing 3D volume $\mathbf{V}_t$ at time t as a weighted sum of shared 3D basis volumes $\left\{\hat{\mathbf{V}}_1, \ldots, \hat{\mathbf{V}}_{R_t}\right\}$ [Equation 2 (i)]. Shared volumes are still not optimal, each requiring independent $\mathrm{M}_r^{XY}, \mathrm{v}_r^Z$; VM-T (Vector, Matrix, and Time) factorization allows the low-rank components to be further shared across shared volumes, improving efficiency [Equation 3]. CANDECOMP Decomposition (CP Decom.) represents 4D volumes using vectors representing individual axes, instead of bi-axial matrices, thus decoupling the axes

[Equation 2 (ii)].

$$\mathbf{V} = \sum_{r=1}^{R_1} \mathbf{M}_r^{XY} \circ \mathbf{v}_r^Z \circ \mathbf{v}_r^1 + \sum_{r=1}^{R_2} \mathbf{M}_r^{XZ} \circ \mathbf{v}_r^Y \circ \mathbf{v}_r^2 + \sum_{r=1}^{R_3} \mathbf{M}_r^{ZY} \circ \mathbf{v}_r^X \circ \mathbf{v}_r^3 \tag{1}$$

$$\text{(i)} \quad \mathbf{V}_t = \sum_{i=1}^{R_t} f(t)_i \cdot \hat{V}_i \qquad \text{(ii)} \quad \mathbf{V}_t = \sum_{r=1}^{R} \mathbf{v}_r^X \circ \mathbf{v}_r^Y \circ \mathbf{v}_r^Z \circ \mathbf{v}_r \cdot \mathbf{f}_r(t) \tag{2}$$

$$\mathbf{V}_t = \sum_{r=1}^{R_1} \mathbf{M}_r^{XY} \circ \mathbf{v}_r^Z \circ \mathbf{v}_r^1 \cdot \mathbf{f}^1(t)_r + \sum_{r=1}^{R_2} \mathbf{M}_r^{XZ} \circ \mathbf{v}_r^Y \circ \mathbf{v}_r^2 \cdot \mathbf{f}^2(t)_r + \sum_{r=1}^{R_3} \mathbf{M}_r^{ZY} \circ \mathbf{v}_r^X \circ \mathbf{v}_r^3 \cdot \mathbf{f}^3(t)_r \tag{3}$$

### 2.3.3 HexPlane Representation

These factorizations decouple the spatial and temporal modeling. However, real-world scenes often involve intertwined spatial and temporal dynamics. This is addressed by the replacement of $\mathbf{v}_r^Z \circ \mathbf{f}^1(t)_r$ in Equation 3 with a joint function of both time and space, represented akin to a piecewise linear function. Achieving this involves bilinear interpolation into a learned tensor with dimensions $\mathbb{Z} \times T \times R1$. The 4D feature volume $V \in \mathbb{R}^{XYZTF}$, after applying the same transformation to similar terms, is represented as:

$$\mathbf{D} = \sum_{r=1}^{R_1} (\mathbf{M_r^{XY}} \cdot \mathbf{M_r^{ZT}} \cdot \mathbf{v_r^1}) + \sum_{r=1}^{R_2} (\mathbf{M_r^{XZ}} \cdot \mathbf{M_r^{YT}} \cdot \mathbf{v_r^2}) + \sum_{r=1}^{R_3} (\mathbf{M_r^{YZ}} \cdot \mathbf{M_r^{XT}} \cdot \mathbf{v_r^3}) \tag{4}$$

Here, each $\mathbf{M_r^{AB}} \in \mathbb{R}^{AB}$ is a learned plane of features.

Alternatively, the model can be represented as the function $D$ mapping $(x, y, z, t)$ to an $F$ dimensional vector as:

$$\mathbf{D(x, y, z, t)} = \left( \mathbf{P}_{xy\bullet}^{XYR_1} \odot \mathbf{P}_{zt\bullet}^{ZTR_1} \right) \mathbf{V}^{R_1F} + \left( \mathbf{P}_{xz\bullet}^{XZR_2} \odot \mathbf{P}_{yt}^{YTR_2} \right) \mathbf{V}^{R_2F} + \left( \mathbf{P}_{yz\bullet}^{YZR_3} \odot \mathbf{P}_{xt\bullet}^{XTR_3} \right) \mathbf{V}^{R_3F} \tag{5}$$

Where $\odot$ is an elementwise product, the superscript of each bold tensor represents its shape, and $\bullet$ in a subscript represents a slice, so each term is a vector-matrix product. $\mathbf{P}^{XYR_1}$ stacks all $\mathbf{M}_r^{XY}$ to a 3D tensor, and $\mathbf{V}^{R_1F}$ stacks all $\mathbf{v}_r^1$ to a 2D tensor; other terms are defined similarly. Coordinates $x, y, z, t$ are real-valued, so subscripts denote bilinear interpolation. We can stack all $\mathbf{V}^{R_iF}$ into $\mathbf{V}^{RF}$ and rewrite the equation as:

$$\left[ \mathbf{P}_{xy\bullet}^{XYR_1} \odot \mathbf{P}_{zt\bullet}^{ZTR_1}; \mathbf{P}_{xz\bullet}^{XZR_2} \odot \mathbf{P}_{yt\bullet}^{YTR_2}; \mathbf{P}_{yz\bullet}^{YZR_3} \odot \mathbf{P}_{xt\bullet}^{XTR_3} \right] \mathbf{V}^{RF} \tag{6}$$

### 2.3.4 Optimization and Regularization

The optimization strategies employed in the original research are utilized, including photometric loss for model training and specific regularizers to address the ill-posed nature of dynamic 3D scene reconstruction. We follow the outlined coarse-to-fine training scheme and incorporate the proposed regularizers, such as Total Variation (TV) loss and depth smooth loss, as shown in Niemeyer et al. (2022), to enhance the quality of the synthesized views and minimize artifacts.

## 3 Experiments

### 3.1 Objective

Our study aims to reproduce the experimental evaluation of HexPlane, an explicit representation proposed for dynamic novel view synthesis. We sought to assess HexPlane's design choices, performance, and efficiency across challenging datasets, compare its outcomes to state-of-the-art methods, and validate its robustness under various conditions. The primary goal was to examine the simplicity, generality, effectiveness, and architectural choices of HexPlane. We begin with reproduction experiments followed by additional analysis and minor improvements over the method.

### 3.2 Datasets

To replicate the original experiments, we closely followed the described methods, concentrating on two primary datasets:

**Plenoptic Video Dataset** (Li et al., 2022): A high-resolution, multi-camera dataset showcasing dynamic content captured with 21 GoPro cameras at 2.7K resolution. This dataset incorporates complex motion and fine details over extended videos to assess HexPlane's representational capacity. The dataset consists of 6 scenes, each captured with 18 synchronized cameras. The cameras were arranged in a circle around the scene so that each camera captured a slightly different view of the scene. Quantitative Metrics on this dataset are provided in table 1.

Table 1: Quantitative Comparisons on Plenoptic Video Dataset
* represents the model with fewer training steps

| Model | Steps | PSNR↑ | | LPIPS v↓ | |
|---|---|---|---|---|---|
| | | Ours | Paper | Ours | Paper |
| HexPlane-all | 650k | 30.247 | 31.705 | 0.101 | 0.075 |
| HexPlane-all* | 100k | 31.244 | 31.569 | 0.094 | 0.089 |

**D-NeRF Dataset** (Pumarola et al., 2020): A monocular video dataset featuring synthetic objects designed to assess the model's performance with extremely sparse observations and its capability to synthesize novel views from monocular videos. This dataset contains 8 scenes on different contexts and objects. The scenes contain objects undergoing rigid, articulated, and non-rigid motions, rendered from various viewpoints at consecutive time steps. Quantitative Metrics on this dataset are provided in table 2.

### 3.3 Comparative Study

Various methods employing explicit representation of such scenes and a lightweight decoder have shown promising results. The most notable works closely related to HexPlanes are K-planes Fridovich-Keil et al. (2023) and 4D-Gaussian Splatting Wu et al. (2023). Here, we discuss the characteristics of these works in the form of a comparative study.

**K-Planes** Similar to HexPlane, K-Planes also factor the 4D spatial-temporal scene into six feature planes. In K-planes, plane features projected on all six planes are multiplied using the Hadamard product. In HexPlane, two fusion mechanisms, multiplication followed by concatenation, are applied to the plane features. K-Planes (explicit) uses a linear feature decoder for RGB and density values with a learned color basis instead of the black-box MLP decoder in Hexplane for regressing RGB values. The K-Planes model employs simultaneous queries through planes of varied spatial resolutions (e.g., 64, 128, 256, and 512) to make the model robust at higher and lower resolutions. Hence, the main difference between the architectures of HexPlane and K-Planes is how these models generate the feature vector from the six projections of the 4-D point(x, y, z, t). It can be observed that both HexPlane and KPlane have similar architectures, resulting in comparable performance metrics.

**4D-Gaussian Splatting** 4D-Gaussian Splatting(4D-GS) uses a novel explicit representation, containing both 3D Gaussians and 4D neural voxels. A decomposed neural voxel encoding algorithm inspired by HexPlane is proposed to build Gaussian features efficiently from 4D neural voxels. Then a lightweight MLP is applied to predict Gaussian deformations at novel timestamps. The 4D-GS framework includes 3D Gaussians 'G' and Gaussian deformation field network 'F.' The Gaussian deformation field network consists of an efficient spatial-temporal structure encoder, 'H,' and a Multi-head Gaussian Deformation Decoder, 'D.' In the spatial-temporal structure encoder 'H,' the neural voxel encoding scheme and a tiny MLP inspired by HexPlanes are used to merge all the features of the input 3D Gaussians. Further in the Multi-head Gaussian Deformation Decoder 'D', taking the encoded features from the encoder 'H' as input, the deformations of the 3D Gaussians are decoded using separate MLPs, and a deformed 3D Gaussian are obtained at timestep 't.' The deformed Gaussians are then splatted to the rendered image.

From a quantitative evaluation of these methods on the D-Nerf dataset 2, we observe that HexPlane has a slightly better PSNR and LPIPS than K-Planes. 4D-GS is an improvement on HexPlane and has the best scores. However, we analyze the base HexPlane model to verify the root model and approach and ensure its reliability as a baseline.

The use of multiple spatial resolutions across the planes helps K-planes achieve robustness at both high and low resolutions, making it well-suited for diverse scene complexities. The use of Gaussian deformations in 4D-GS allows for detailed and dynamic scene representation, which might explain its superior performance on metrics like PSNR and LPIPS as compared to the other two models. The base model, HexPlane, while sharing similar architecture to K-planes, utilizes two fusion mechanisms—multiplication followed by concatenation—which may not be as efficient as the single-step Hadamard product used in K-planes.

Table 2: 4D-Gaussian Splatting and K-Planes Benchmark on D-Nerf Dataset

| Scene | Models | PSNR↑ | SSIM↑ | LPIPS v↓ |
|---|---|---|---|---|
| Bouncing Balls | Gaussian Splatting | 40.348 | 0.994 | 0.030 |
| | K-Planes | 40.063 | 0.994 | 0.033 |
| | Hex-Plane | 40.463 | 0.993 | 0.029 |
| Hell Warrior | Gaussian Splatting | 28.983 | 0.974 | 0.047 |
| | K-Planes | 24.681 | 0.954 | 0.082 |
| | Hex-Plane | 24.338 | 0.944 | 0.074 |
| Hook | Gaussian Splatting | 32.975 | 0.977 | 0.031 |
| | K-Planes | 28.130 | 0.959 | 0.067 |
| | Hex-Plane | 28.262 | 0.955 | 0.053 |
| Jumping Jacks | Gaussian Splatting | 35.505 | 0.986 | 0.025 |
| | K-Planes | 31.410 | 0.971 | 0.057 |
| | Hex-Plane | 31.710 | 0.974 | 0.036 |
| Lego | Gaussian Splatting | 25.150 | 0.938 | 0.062 |
| | K-Planes | 25.412 | 0.941 | 0.043 |
| | Hex-Plane | 25.144 | 0.940 | 0.042 |
| Mutant | Gaussian Splatting | 37.197 | 0.987 | 0.022 |
| | K-Planes | 32.582 | 0.987 | 0.046 |
| | Hex-Plane | 33.686 | 0.980 | 0.025 |
| Stand Up | Gaussian Splatting | 37.410 | 0.989 | 0.018 |
| | K-Planes | 33.099 | 0.923 | 0.033 |
| | Hex-Plane | 34.121 | 0.983 | 0.020 |
| T Rex | Gaussian Splatting | 33.765 | 0.984 | 0.028 |
| | K-Planes | 31.270 | 0.965 | 0.048 |
| | Hex-Plane | 30.953 | 0.975 | 0.028 |
| Average | Gaussian Splatting | 33.917 | 0.979 | 0.033 |
| | K-Planes | 30.788 | 0.962 | 0.051 |
| | Hex-Plane | 31.084 | 0.968 | 0.038 |

### 3.4 Factorization Design

Dynamic 3D scenes can naturally be modeled as a 4D volume. Still, significant challenges are encountered, including its high memory usage and sparse observations due to the need for multiple frames per timestamp. Several potential factorization techniques are proposed in the original study, including decomposing the large original volume into smaller latent outputs to address this issue. Subsequently, we conduct assessments employing the diverse factorization methodologies recommended on the D-NeRF dataset, as shown in 3. From the empirical results of these variations, the choice of HexPlane representation is justified.

It is observed that the HexPlane factorization technique performs considerably better than others, with significantly reduced training times. This demonstrates that HexPlane strikes a balance between shared volumes, fewer parameters, and the coupling of axes.

Table 3: Quantitative Results for Different Factorizations
R = 16 for Volume Basis, and R = 48 for the rest

| Model | PSNR↑ | | SSIM↑ | | LPIPS v↓ | | Training Time |
|---|---|---|---|---|---|---|---|
| | Ours | Paper | Ours | Paper | Ours | Paper | Ours |
| Volume Basis | 29.343 | 30.631 | 0.923 | 0.967 | 0.049 | 0.042 | 30m |
| VM-T | 31.598 | 30.657 | 0.921 | 0.965 | 0.031 | 0.048 | 17m |
| CP Decom. | 29.538 | 28.370 | 0.922 | 0.942 | 0.061 | 0.083 | 12m |
| HexPlane | 31.084 | 31.042 | 0.968 | 0.970 | 0.025 | 0.039 | 12m |

Table 4: Ablations on Feature Plane Designs

| Model | PSNR↑ | | SSIM↑ | | LPIPS v↓ | |
|---|---|---|---|---|---|---|
| | Ours | Paper | Ours | Paper | Ours | Paper |
| Spatial Planes | 20.296 | 20.369 | 0.853 | 0.879 | 0.176 | 0.148 |
| Spatial-Temporal Planes | 20.323 | 21.112 | 0.934 | 0.879 | 0.123 | 0.148 |
| DoublePlane (XY-ZT) | 30.145 | 30.370 | 0.956 | 0.961 | 0.043 | 0.054 |
| HexPlane-Swap | 26.224 | 28.562 | 0.931 | 0.954 | 0.072 | 0.056 |
| HexPlane | 31.084 | 31.042 | 0.968 | 0.970 | 0.025 | 0.039 |

## 3.5 Feature Plane Designs

The hexagonal plane demonstrates excellent symmetry due to its inclusion of all pairs of coordinate axes, both spatial planes $P_{XY}, P_{YZ}, P_{ZX}$ and spatial-temporal planes $P_{XT}, P_{YT}, P_{ZT}$. Evaluation of the model's performance on different sets of planes by breaking this symmetry is provided i,n table 4.

As the table demonstrates, neither Spatial Planes nor Spatial-Temporal Planes alone could represent dynamic scenes, highlighting the need to incorporate time and space for adequate representation. The DoublePlane consists of solely one set of paired planes, namely $P_{XY}$ and $P_{ZT}$. On the other hand, the HexPlane-Swap arranges planes in groups where axes are duplicated, such as $P_{XY}$ and $P_{XT}$. Table 4 also shows the performance for these choices of sets of planes.

Spatial-temporal planes offer distinct advantages, particularly in their ability to effectively model motion within HexPlane using a modest basis number R. This results in enhanced efficiency compared to alternative approaches. As R increases for representation purposes, improved outcomes are achieved, albeit with increased computational requirements.

## 3.6 Feature-Fusion Methods

This section explores HexPlane's key attributes, emphasizing its notable performance in diverse design choices. HexPlane employs various feature fusion mechanisms, including Multiply-Concat, Sum-Multiply, and Multiply-Sum. A comprehensive analysis of fusion ablations is presented in Table 5, specifically focusing on Fusion-One and Fusion-Two. This involves exploring combinations of fusion methods such as Concat, Sum, and Multiply. Multiply-Concat produces the best results, while Sum-Sum or Sum-Concat provides the worst results. Additionally, opacity features are sampled as 8-dimensional vectors from HexPlane and regressed using another MLP.

It is crucial to note the significance of weight initializations for feature planes in different fusion designs. For instance, Multiply-Multiply and Concat-Multiply require Gaussian noise initialization (mean 0.5, scale 0.9), while others follow a mean 0.0 and scale 0.1 initialization. A scale of 0.1 or 0.9 implies that most features being multiplied are lesser than 1. This observation is justified because the multiplication of values lesser than one leads to even smaller values, thus requiring a greater Gaussian noise initialization.

Table 5: Ablations on Feature Fusion Designs
* represents results of the original paper

| FusionOne | FusionTwo | PSNR↑ | SSIM↑ | LPIPS a↓ |
|---|---|---|---|---|
| Multiply | Concat | 30.513 | 0.934 | 0.043 |
| | * | 31.042 | 0.968 | 0.039 |
| | Sum | 31.400 | 0.964 | 0.033 |
| | * | 31.023 | 0.967 | 0.039 |
| | Multiply | 30.153 | 0.963 | 0.052 |
| | * | 30.345 | 0.966 | 0.041 |
| Sum | Concat | 24.104 | 0.924 | 0.100 |
| | * | 25.428 | 0.931 | 0.084 |
| | Sum | 24.046 | 0.913 | 0.122 |
| | * | 25.227 | 0.928 | 0.090 |
| | Multiply | 29.032 | 0.980 | 0.054 |
| | * | 30.585 | 0.965 | 0.044 |

Table 6: Dynamic View Synthesis without MLPs
* represents results of the original paper

| Model | PSNR | SSIM | LPIPS-a | LPIPS-v | Training Time |
|---|---|---|---|---|---|
| Hexplane | 31.084 | 0.968 | 0.025 | 0.038 | 11m 22s |
| HexPlane* | 31.042 | 0.968 | | 0.039 | 11m 30s |
| Hexplane-SH | 29.169 | 0.955 | 0.036 | 0.053 | 10m 19s |
| HexPlane-SH* | 29.284 | 0.952 | | 0.056 | 10m 42s |

### 3.7 Spherical Harmonics

In pursuing MLP-free designs, HexPlane-SH is utilized for Dynamic View Synthesis on the D-NeRF dataset. This explicit model uses spherical harmonics (SH) coefficients as appearance features. By directly regressing RGB values from these SH coefficients, HexPlane-SH achieves comparable results to Hexplane with MLP, as shown in the tables 6 and 9. Spherical Harmonics Color Decoding is explored as an alternative to MLP-based color regression, wherein SH coefficients are computed directly from HexPlanes and subsequently decoded to RGBs using view directions. Spherical Harmonics are evaluated at unit directions, leveraging hardcoded SH polynomials up to degree 4. SH values are computed based on input coefficients and directions. Although this presents a marginal reduction in quality, it has faster rendering speeds.

### 3.8 HexPlane Slim

A Slim version of the Hexplane model is evaluated, which directly outputs density values for rendering rather than using an MLP to convert a density feature vector from Hexplane to a single value output density. When *density-dim* is set to 1 and *DensityMode* is set to "plain", densities are taken directly from the HexPlane without MLPs. When *density-dim* is set to 8 and *DensityMode* is set to "general-MLP", densities undergo MLP processing to predict scalar values as per the original implementation. We observe a slightly better performance with the MLP's, than the slim version. This shows that the MLP is needed as a functional approximator to regress the outputs values from the feature plane outputs. Expecting the feature planes to directly output values appears to overload the representation network.

### 3.9 iPhone Dataset

Due to several key factors, the iPhone dataset proposed in Gao et al. (2022) poses unique challenges compared to general Dynamic Real Datasets. Typical datasets like D-NeRF and Plenoptic either contain frames that teleport between multiple camera viewpoints at consecutive time steps, impractical to capture from a single camera, or depict quasi-static scenes, which do not represent real-life dynamics. The iPhone captures diverse

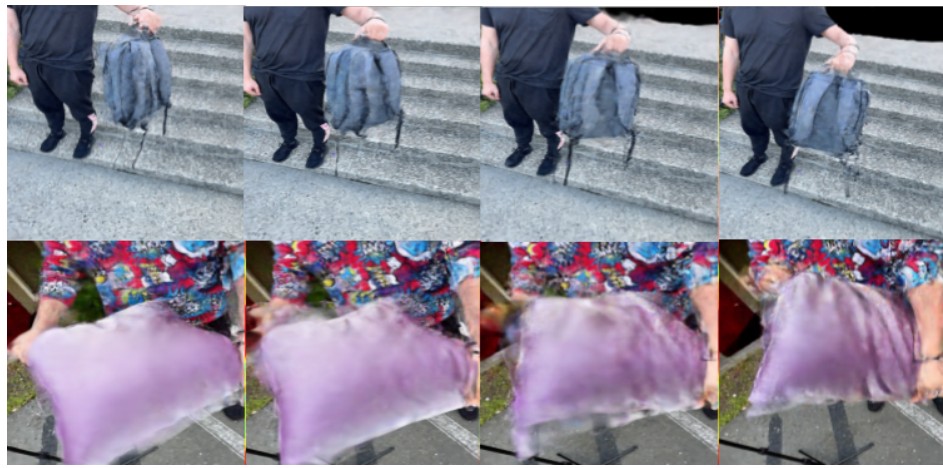

Figure 2: HexPlane is evaluated on iPhone-captured casual videos, demonstrating dynamic novel view synthesis across various time steps and viewpoints.

Table 7: Performance Metrics for iPhone Dataset

| Scene | PSNR_test | SSIM | LPIPS_a | LPIPS_v |
|---|---|---|---|---|
| apple | 15.426 | 0.320 | 0.778 | 0.685 |
| block | 15.172 | 0.315 | 0.785 | 0.690 |
| space-out | 14.940 | 0.462 | 0.681 | 0.668 |
| backpack | 22.650 | 0.644 | 0.368 | 0.376 |
| pillow | 18.674 | 0.586 | 0.354 | 0.432 |
| wheel | 12.142 | 0.225 | 0.601 | 0.616 |
| teddy | 12.730 | 0.242 | 0.784 | 0.733 |
| Average | 15.867 | 0.399 | 0.622 | 0.600 |

real-life scenarios with non-repetitive motions, interactions, and occlusions, making it more challenging than the controlled environments in these datasets.

The original paper needs quantitative metrics pertaining to the iPhone dataset, as the Hexplane author refrains from providing such data. Instead, the author directs attention to GitHub, highlighting an issue with the functionality of the data loader, suggesting that it does not operate as originally intended. We fixed it and evaluated the model on 7 scenes for a more robust evaluation of the HexPlane model, as shown in Table 7. We observe that the values are primarily different from those of the other datasets. This is expected, as demonstrated by Gao et al. (2022) on other existing methods for dynamic scenes.

### 3.10 Space Only - Time Only Visualisation

In cases where the scenes are static, the model leverages features solely from the space planes, resulting in efficient compression benefits. Moreover, the model allows for tracking temporal changes by visualizing elements in the time-space planes that deviate from 1. This means that changes occurring over time are explicitly captured. Therefore, the visualization of either space or time independently justifies the utilization of both spatial and temporal planes, underscoring the model's versatility in providing insights into the scene's static and dynamic aspects.

The distinct separation of space-only and space-time planes could demonstrate the model's interpretability. In cases where the scenes are static, the model leverages features solely from the space planes, resulting in efficient compression benefits. This approach facilitates the identification and concise representation of static regions. Moreover, the model allows tracking temporal changes by visualizing elements in the time-

space planes alone. This means that changes occurring over time are explicitly captured. Therefore, the visualization of either space or time independently, as in 3.10, justifies the utilization of both spatial and temporal planes, underscoring the model's versatility in providing insights into both static and dynamic aspects of the scene.

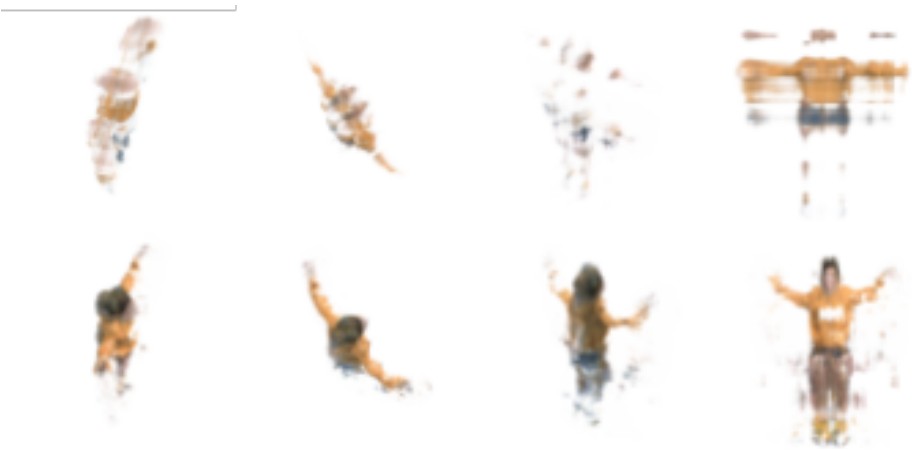

Figure 3: We visualize the space and time planes individually by setting one of the planes to unity. The first row depicts the "space-only" planes, while the latter is the corresponding "time-only" planes. It can be easily observed that both complement each other in all four results.

### 3.11 Temporal Smoothness Loss

Inspired by (Fridovich-Keil et al., 2023), we apply the temporal smoothness regularization exclusively to the temporal dimension of our space-time planes. The pursuit of temporal smoothness in space-time planes plays a pivotal role in refining the visual coherence of dynamic scenes within video processing.

We promote smooth motion by employing an ID Laplacian filter, targeting the penalization of abrupt "acceleration" over time. The filter operationalizes temporal smoothness by calculating the difference between adjacent frames to obtain a 'first difference' across the temporal dimension. It then computes the 'second difference' by finding the difference between consecutive first differences. The measure of temporal smoothness is the L2 norm of these second differences squared.

$$L_{smooth} = \frac{1}{|C|n^2} \sum_{c,i,t} ||P_c^{i,t-1} - 2P_c^{i,t} + P_c^{i,t+1}||_2^2 + \lambda_{\text{reg}} L_{\text{reg}} \tag{7}$$

The resulting loss value serves as a quantification of temporal smoothness within the space-time planes. By systematically tuning hyperparameters, we identified optimal values for lambda, as presented in the accompanying table 8. Notably, the PSNR demonstrates an increasing trend with the elevation of lambda until reaching 0.01. Beyond this threshold, the PSNR decreases, indicating that 0.01 serves as the optimal value. However, the marginal improvement from the previous value (0.001) suggests that temporal loss regularization has a limited impact on the overall results.

This regularization improved the model's stability, reducing sudden accelerations and eliminating abrupt jerks and unnatural jittering in movements. We recognize that such minor adjustments are essential to enhance the output quality.

### 3.12 Limitations

Our replication efforts highlight a few limitations inherent to the HexPlane methodology:

Table 8: Temporal Smoothness Regularization

| Time Smoothness Weight($\lambda$) | PSNR |
|---|---|
| 0 | 31.08 |
| 0.001 | 31.51 |
| 0.01 | 31.63 |
| 0.1 | 30.83 |
| 1 | 29.19 |
| 10 | 29.19 |

Performance with Sparse Observations: Unlike methods that leverage deformation and canonical fields for dynamic 3D scene representation, HexPlane's reliance on a basis-sharing mechanism, though innovative, must be more robust in scenarios with highly sparse data. This is a critical area where its performance is notably impacted.

Artifacts and Regularization Needs: Consistent with observations in the original HexPlane study, our replication process confirmed the occurrence of artifacts, such as color jittering in synthesized results. This underscores the necessity for more robust regularization strategies to mitigate these issues and enhance overall output quality.

### 3.13 Future Directions

Building on the foundational work of HexPlane, we propose several avenues for future research aimed at overcoming these limitations and expanding the model's applicability:

Enhanced Regularization Techniques: The development of specialized spacetime regularizations and the adoption of additional loss functions, such as optical flow loss, may provide pathways to reduce artifacts and improve the fidelity of synthesized scenes.

Basis Variation for Long Videos: Tailoring the basis representation to accommodate variations across different video segments could yield more accurate and dynamic scene representations, addressing one of the core limitations noted in our replication effort.

Utilization of Category-Specific Priors: Exploring the combination of HexPlane with category-specific models, such as 3DMM or SMPL, could offer targeted enhancements for specific scene types, thereby expanding HexPlane's versatility and accuracy in diverse applications.

## 4 Conclusion

Our replication affirms the original claims, demonstrating HexPlane's ability to achieve comparable or superior synthesis quality for dynamic novel view synthesis and notable accelerations exceeding hundreds of times compared to implicit representations.

Our efforts involved verifying the claims and architectural choices, ensuring the robustness and reliability of the HexPlane framework as a baseline. The reproduced findings confirm its potential to revolutionize dynamic scene representation without introducing deformation, category-specific priors, or specific tricks. We contribute the code for additional regularisation on Temporal Smoothness.

In conclusion, our reproduction of the HexPlane paper validates its efficacy and opens doors for future research opportunities and applications in the broader field of 3D scene processing.

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

# A   Appendix

Table 9: Performance Metrics for Different Scenes for Spherical Harmonics

| Scene | PSNR | SSIM | LPIPS-a | LPIPS-v |
|---|---|---|---|---|
| Stand Up | 32.059 | 0.973 | 0.017 | 0.029 |
| Hook | 26.731 | 0.940 | 0.044 | 0.068 |
| Bouncing Ball | 36.768 | 0.985 | 0.012 | 0.054 |
| Hell Warrior | 21.135 | 0.896 | 0.092 | 0.113 |
| Lego | 24.929 | 0.934 | 0.038 | 0.051 |
| Jumping Jacks | 29.846 | 0.966 | 0.032 | 0.046 |
| Mutant | 32.878 | 0.977 | 0.020 | 0.029 |
| T-Rex | 29.007 | 0.970 | 0.030 | 0.032 |
| Average | 29.169 | 0.955 | 0.036 | 0.053 |

Table 10: Performance Metrics for HexPlane_Slim on D-NeRF Dataset on different scenes.

| Scenes | PSNR | SSIM | MSSIM | LPIPS-a | LPIPS_v |
|---|---|---|---|---|---|
| Hell Warrior | 24.514 | 0.944 | 0.968 | 0.049 | 0.074 |
| Mutant | 33.627 | 0.980 | 0.995 | 0.018 | 0.026 |
| Hook | 28.662 | 0.957 | 0.983 | 0.032 | 0.051 |
| Bouncing Balls | 39.634 | 0.991 | 0.994 | 0.008 | 0.032 |
| Lego | 25.124 | 0.939 | 0.961 | 0.033 | 0.044 |
| T-Rex | 30.653 | 0.975 | 0.986 | 0.026 | 0.028 |
| Stand Up | 34.382 | 0.984 | 0.995 | 0.013 | 0.020 |
| Jumping Jacks | 31.217 | 0.973 | 0.984 | 0.027 | 0.039 |
| Average | 30.977 | 0.968 | 0.984 | 0.026 | 0.039 |

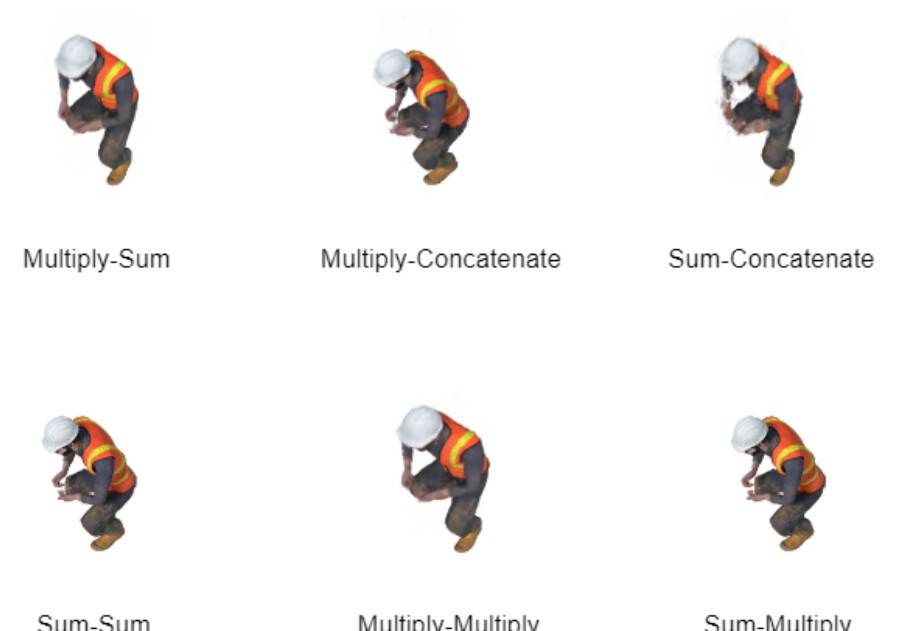

Figure 4: Different fusion designs

