# OpenReview forum: "Exploring Explicit Representations in 4D: A Comparative Analysis with HexPlane"
_TMLR — Rejected by TMLR_

### Review · Reviewer_RcPa · 2024-03-25

**Summary Of Contributions:**

1). This paper replicates and confirms the claims of Hexplane from Cao and Johnson (2023) through a detailed examination of architectural and design choices. 2). This paper evaluate Hexplane's robustness and superior efficacy by testing on novel datasets and comparing with recent methodologies like Gaussian Splatting and K-Planes; 3). This paper improves Hexplane performance by incorporating temporal smoothness regularization.

**Audience:**

Yes

**Claims And Evidence:**

Yes

**Requested Changes:**

1. Highlight the actual contribution and novelty of this paper in this research field.
2. It is not recommended to directly use the Figure 1 from origin Hexplane paper.

**Strengths And Weaknesses:**

Strength:
1. Providing a detailed analysis and conduct thorough ablation study of Hexplane architectural and design choices.
2. Comparing different baselines on variety datasets to showcase the robustness of Hexplane.
3. Improving the origin Hexplane by incorporating  additional temporal smoothness regularization.

Weakness:
1. Insufficient academic contributions and novelty. Despite the thorough examination of Hexplane's architectural and design elements, this paper essentially replicates the original Hexplane method without offering substantial academic contribution.
2. The introduction of the temporal smoothness regularization method falls short of constituting a significant contribution.
3. Sorry to say that but this paper is more like a technical report rather than a serious journal paper.

---

> ### Author Response · Authors · 2024-05-01
>
> > Insufficient academic contributions and novelty. Despite the thorough examination of Hexplane's architectural and design elements, this paper essentially replicates the original Hexplane method without offering substantial academic contribution.
>
> Thank you for your feedback regarding the academic contributions of our paper. The primary aim of our research was to rigorously validate the Hexplane framework, which is becoming a fundamental baseline in studies involving explicit representation in 4D spaces. Given the rising prominence of Hexplane, particularly with subsequent methodologies like 4D Gaussian Splatting building upon it, our comprehensive verification of Hexplane's components and architectural decisions is crucial for ensuring its reliability as a baseline.
> We have extended our analysis by benchmarking Hexplane against contemporary methodologies such as K-Planes and 4D Gaussian Splatting, confirming its robustness and suitability for further research. We will clarify these points in our manuscript, emphasizing the importance of validating such a foundational tool in the field and detailing how our findings support and enhance ongoing and future research.
>
> > The introduction of the temporal smoothness regularization method falls short of constituting a significant contribution.
>
> We appreciate your insights on the inclusion of temporal smoothness regularization in our study. While this enhancement was not intended as a novel contribution, it serves as a practical experimentation to improve model performance in dynamic scenarios. Our results demonstrate that integrating temporal smoothness into Hexplane significantly boosts its stability and accuracy when processing temporal data.
> In the revised manuscript, we better articulate the experimental nature of this regularization technique and its practical benefits.
>
> > Sorry to say that but this paper is more like a technical report rather than a serious journal paper.
>
> Thank you for pointing out the perceived technical report-like nature of our manuscript. In response, we have thoroughly revised the document to elevate its scholarly impact. We have enhanced the narrative by integrating a more profound discussion of the implications of our findings.
>
> > Highlight the actual contribution and novelty of this paper in this research field.
>
> In our revision, we have refined our introduction section to explicitly outline the contributions of our research, particularly emphasizing the necessity and impact of validating Hexplane as a reliable baseline.
>
> > It is not recommended to directly use the Figure 1 from origin Hexplane paper.
>
> We acknowledge the reviewer's concern regarding our use of figures from the original Hexplane paper. We have replaced the figure.

---

### Review · Reviewer_TAbi · 2024-03-29

**Summary Of Contributions:**

The paper provides an analysis and discussion focused work. Building upon Cao et al.'s HexPlane paper (from CVPR'23), it explores and discusses architectural choices and other related design elements through extensive comparative experiments, validating their robustness across different datasets and benchmarks. Additionally, it introduces a regularization method to enhance performance.

**Audience:**

Yes

**Broader Impact Concerns:**

None.

**Claims And Evidence:**

Yes

**Requested Changes:**

See above comments for details.

**Strengths And Weaknesses:**

Strengths:

1. The exploration direction of the paper is intriguing, particularly in the current stage of development transitioning from 3D to 4D modeling, delving into 4D representations is a promising direction.

2. The experimental design is comprehensive, covering essential aspects of HexPlane in terms of architectures and other design choices, with authors providing many quantitative comparative experimental results.

3. The writing of the paper is relatively clear, making it easy to read and comprehend.



Weaknesses:

1. The motivation for selecting HexPlane for in-depth analysis is not sufficiently explained.

2. There are too few qualitative results, making it difficult to intuitively grasp the differences between different designs.

3. Although the authors provide many comparative experimental results, the analysis and discussion of the results are not sufficiently thorough. For example, the optimal choice of feature fusion methods varies depending on the task or network architecture, and the authors mention that feature fusion methods are a key factor in HexPlane's notable performance. So, what are the preferences for spatial features and temporal features when dealing with 4D data? Why is this the case? The above is just a simple example; similar, more in-depth discussions can inspire the community and further enhance the value of this paper.

4. For 4D scene modeling, many previous works have explored representation methods, so discussions on this problem should not be limited to works related to novel view rendering or within the NeRF framework. To name a few, Occupancy Flow [1], 4D Compositional Representation [2], LoRD [3].

5. The figures in the paper are somewhat blurry; increasing the resolution is recommended.


Minors:

There is an figure index error in the first paragraph of section 4.4.


[1] Niemeyer M, Mescheder L, Oechsle M, et al. Occupancy flow: 4d reconstruction by learning particle dynamics.

[2] Jiang B, Zhang Y, Wei X, et al. Learning compositional representation for 4d captures with neural ode.

[3] Jiang B, Ren X, Dou M, et al. Lord: Local 4d implicit representation for high-fidelity dynamic human modeling.

---

> ### Author Response · Authors · 2024-05-01
>
> > The motivation for selecting HexPlane for in-depth analysis is not sufficiently explained.
>
> Thank you for highlighting the need for a clearer motivation for our selection of HexPlane for in-depth analysis. The choice was driven by HexPlane’s increasing adoption as a fundamental baseline in studies involving explicit representation in 4D spaces. Given the rising prominence of Hexplane, particularly with subsequent methodologies like 4D Gaussian Splatting building upon it, our comprehensive verification of Hexplane's components and architectural decisions is crucial for ensuring its reliability as a baseline. We have enhanced our manuscript by providing a more detailed discussion on why HexPlane was chosen, including its recent applications and relevance to current technological shifts in the field.
>
> > There are too few qualitative results, making it difficult to intuitively grasp the differences between different designs.
>
> We appreciate your comment on the scarcity of qualitative results in our paper. We have included an additional visual comparison in appendix, to better illustrate the differences of various design choices within HexPlane. However, the qualitative figures are nearly indistinguishable from one another, due to the slight variations in performance across different design choices.
>
> > Although the authors provide many comparative experimental results, the analysis and discussion of the results are not sufficiently thorough.
>
> Thank you for pointing out the need for deeper analysis in our discussion of results. We have updated our manuscript to provide clearer insights and a stronger interpretation to support our experimental findings.
>
> > For 4D scene modeling, many previous works have explored representation methods, so discussions on this problem should not be limited to works related to novel view rendering or within the NeRF framework. To name a few, Occupancy Flow [1], 4D Compositional Representation [2], LoRD [3].
>
> Thank you for your suggestion to broaden the scope of our discussion on 4D representation methods. However, our focus on NeRF-based methods stems from their established applicability to the specific challenges of real-world environments, as opposed to controlled settings. Established methods like 'Occupancy Flow', '4D Compositional Representation', and 'LoRD' have primarily been tested on datasets such as point clouds or meshes featuring single subjects with a defined structure. These methods often emphasize shape over visual rendering—with the exception of 'LoRD'.
>
> Our aim is to apply HexPlane to more complex scenarios found in real-life applications, such as the plenoptic dataset, which features multi-subject crowded scenes captured via camera images. While it would be intriguing to adapt and compare these established methods to the dnerf dataset, and potentially even to the plenoptic dataset, we aim to do so in the future due to limited rebuttal time period.
>
> We also look forward to the development of new non-NeRF-based methods that can effectively represent real world complex dynamic scenes expanding from single subject scenes to real world complex dynamic scenes like plenoptic using real-life camera inputs, expanding the possibilities for 4D scene modeling.
>
> > The figures in the paper are somewhat blurry; increasing the resolution is recommended.
>
> We thank you for pointing out the issue with the figures’ clarity. We will replace or enhance all figures in question

---

### Review · Reviewer_hhJ7 · 2024-03-31

**Summary Of Contributions:**

This paper studies HexPlane, a prior art for reconstructing 3D dynamic scenes. The paper revisits some of the design decisions in HexPlane and conducts several ablation studies on them. In addition, it compares HexPlane with two other methods: 4D Gaussian splatting and K-planes. The paper discusses their performances on different datasets.

The paper mostly evaluates a prior art to validate its design decision and efficacy. Therefore, I am unsure if I see enough technical novelty or contribution over the existing HexPlane work. The empirical results from the experiments might be interesting to people in the community, though.

**Audience:**

No

**Broader Impact Concerns:**

I don't see any.

**Claims And Evidence:**

No

**Requested Changes:**

- I suggest rewriting the paper to articulate its motivation (Abstract/Introduction). I also suggest the paper examines its arguments carefully and provides a more balanced and comprehensive view of prior arts (Related work).
- Please clarify the contributions from prior art and from this work in the Method and Experiment sections.
- Analyze the performance difference between HexPlane and the two chosen baselines (K-plane and 4D Gaussian splatting). It would be helpful if the paper could reveal some insights here, e.g., what components or steps make one method outperform another?
- Analyze these methods on more new datasets not discussed in the original HexPlane. The iPhone images (Fig. 2) seem to be a good starting point. A more comprehensive analysis will make this paper stronger. For example, it could be interesting to consider various properties of the data (static, dynamic, rigid, soft, convex, concave, occlusion, organic surfaces, CAD designs, etc.) and analyze how they may affect the performance.

**Strengths And Weaknesses:**

**Strengths**

It won’t hurt to have an extra pair of eyes to sanity-check the design decision and the efficacy of a previous paper, but I am not confident that this is a strong contribution. In addition, the limitations (Sec. 4.12) and future directions (Sec. 4.13) might be interesting.

**Weaknesses**

- The paper’s writing can be improved. There are many grammatical errors, and the paper would really benefit from careful proofreading.

- What concerns me more is that I feel the abstract and the introduction didn’t do a good job of presenting a well-grounded story. A number of arguments in the paper present incomplete information that may be misinterpreted by users unfamiliar with the field. For example:
1. The first sentence in the introduction “Current methods of reconstructing dynamic scenes…build upon NeRF…” ignores methods that are not NeRF-based.
2. The paper frequently uses the term “the authors”. Sometimes it is unclear whether it refers to the authors of this submission or a referenced paper. This makes the distinction between what this paper and other prior works contribute less clear.
3. There is little connection between the introduction and its summary (“To summarize”), and I find it difficult to understand the paper’s motivation and significance from its introduction.
4. Sec. 2.1 ends with works on static scenes and leaves readers with the impression that NeRF/Neural scene representation has not been explored for 3D dynamic scenes. This is an incomplete argument. Some sentences at the beginning of Sec. 2.2 could have been merged here to avoid misinterpreting the state-of-the-art NeRF works.

I stopped tracking them at the end of the related work, but similar issues remain in the rest of the paper.

- I am worried about the whole Method section, as it largely overlaps with the technical method described in HexPlane. In particular, Fig. 1 is a copy from HexPlane. Although the paper makes it clear that it replicates HexPlane, the presentation of this Method section still disturbs me. In my opinion, naming it a Background section would be better, as “Background” would properly indicate that the section describes a prior art.

- I don’t quite understand the experiment objective (Sec. 4.1), as HexPlane is a peer-reviewed paper and has released its source code on GitHub for quite a while. It looks like HexPlane also evaluated its method on the two datasets described in Sec. 4.2 of this submission.

- The comparative study (Sec. 4.3) could benefit from a more in-depth discussion of the performance difference between these methods. The last paragraph reports the statistics/numerical facts but does not provide an analysis.

- I also don’t get the motivation for the experiments in Secs. 4.4-4.7. How are they different from the ablation study reported in HexPlane? I am worried about the similarity here.

---

> ### Author Response · Authors · 2024-05-01
>
> Thank you for your detailed feedback and the insights provided.
>
> > The paper’s writing can be improved. There are many grammatical errors, and the paper would really benefit from careful proofreading.
>
> We have corrected the grammatical issues and rewritten the paper for improved clarity and coherence.
>
> > What concerns me more is that I feel the abstract and the introduction didn’t do a good job of presenting a well-grounded story.
>
> We have revised the abstract and introduction to be well representative of our research, key insights, and its significance.
>
> > The first sentence in the introduction “Current methods of reconstructing dynamic scenes…build upon NeRF…” ignores methods that are not NeRF-based.
>
> We have updated the introduction to acknowledge the variety of methods used for reconstructing dynamic scenes, including those not based on NeRF, providing a more comprehensive overview of the field. However, we restricted our analysis to nerf based methods, since existing non-nerf methods like [1], [2], [3] are primarily for single-subject, clear background scenes with defined structures, unlike nerf based methods which test on multi-subject crowded scenes captured using camera images, to stay more relevant to current research demands.
>
> > The paper frequently uses the term “the authors”. Sometimes it is unclear whether it refers to the authors of this submission or a referenced paper. This makes the distinction between what this paper and other prior works contribute less clear.
>
> We have clarified the usage of 'the authors' throughout the paper to distinctly identify when we are referring to our own team versus original authors of hexplane.
>
> > There is little connection between the introduction and its summary (“To summarize”)
>
> We have rewritten the introduction and summary to clearly convey the motivation and significance of our study.
>
> > Sec. 2.1 ends with works on static scenes and leaves readers with the impression that NeRF/Neural scene representation has not been explored for 3D dynamic scenes.
>
> We have restructured the entire "Background and Related works" section.
>
> > In my opinion, naming it a Background section would be better, as “Background” would properly indicate that the section describes a prior art.
>
> We have restructured the Hexplane prior art section under 'Background and Related works'. We have done a complete revision delivering a different perspective and interpretation of the original method.
>
> > I don’t quite understand the experiment objective (Sec. 4.1)
>
> We replicated the experiments from HexPlane as an initial step to ensure reproducibility before we extend it further in our subsequent analyses.
>
> > The comparative study (Sec. 4.3) could benefit from a more in-depth discussion of the performance difference between these methods. The last paragraph reports the statistics/numerical facts but does not provide an analysis.
>
> We have enhanced Section 4.3 to include a deeper analysis of the performance differences. It now provides a more thorough interpretation of the quantitative results, discussing the differences with HexPlane.
>
> > I also don’t get the motivation for the experiments in Secs. 4.4-4.7.
>
> We appreciate your concern regarding the potential redundancy in Sections 4.4-4.7. Code was not supplied for these architectural variations. We contribute the code and have refined the text to clearly demonstrate the implications and insights from these results.
>
> > I suggest rewriting the paper to articulate its motivation (Abstract/Introduction). I also suggest the paper examines its arguments carefully and provides a more balanced and comprehensive view of prior arts (Related work).
>
> We have reviewed and refined the entire manuscript, particularly the related work section, to maintain consistency and prevent any potential misinterpretation.
>
> > Analyze these methods on more new datasets not discussed in the original HexPlane. The iPhone images (Fig. 2) seem to be a good starting point.
>
> Thank you for your suggestion to explore diverse datasets. We agree on the importance of such analysis but, we have encountered practical limitations in accessing datasets with such diversity, particularly for dynamic scenes, suitable for NeRF setting, particularly within the tight timelines of this rebuttal period. We plan to pursue more varied datasets in future work to further validate and enhance our findings.
>
> [1] Niemeyer M, Mescheder L, Oechsle M, et al. Occupancy flow: 4d reconstruction by learning particle dynamics.
>
> [2] Jiang B, Zhang Y, Wei X, et al. Learning compositional representation for 4d captures with neural ode.
>
> [3] Jiang B, Ren X, Dou M, et al. Lord: Local 4d implicit representation for high-fidelity dynamic human modeling.

---

### Decision · Action_Editor_T6EU · 2024-05-15

**Recommendation:** Reject

**Comment:**

Please refer to the comments above.

**Audience:**

This paper may be interested to the audience of machine learning community.

**Claims And Evidence:**

The paper offers a comprehensive analysis of Hexplane, revisiting its design choices and evaluating its performance against new datasets and recent methodologies. While the introduction of temporal smoothness regularization aims to enhance Hexplane, the paper suffers from several weaknesses:

1, Overlap with Original Hexplane Work: There is significant overlap with the original Hexplane study, making it difficult to discern new insights or advancements provided by this paper.

2, Writing issue for "Technical Report Rather Than Academic Paper": The paper reads more like a technical report than a substantial journal article, lacking the depth and innovation typically expected in academic research. It may potentially be worried about the academic integrity of this paper.

3, Use of Figures from Original Hexplane Paper: The direct use of figures from the original Hexplane paper is not recommended, as it further diminishes the perceived originality of the work.


Overall, while the empirical results and comparative analyses may interest the community, the paper does not present enough technical content or significant academic contributions to warrant acceptance. All the reviewers suggested the rejection.